# On the Precipice of Extinction: Genetic Data in the Conservation Management of In Situ and Ex Situ Collections of the Critically Endangered *Muehlenbeckia tuggeranong* (Tuggeranong Lignum)

**DOI:** 10.3390/plants14121812

**Published:** 2025-06-12

**Authors:** Isobel Walcott, Angela Lanspeary, Foyez Shams, Peter Bredell, Emma Cook, William Higgisson

**Affiliations:** 1Centre for Applied Water Science, Faculty of Science and Technology, University of Canberra, Bruce, ACT 2617, Australia; angela.lanspeary@canberra.edu.au (A.L.); will.higgisson@canberra.edu.au (W.H.); 2Centre for Conservation Ecology and Genetics, Institute for Applied Ecology, Faculty of Science and Technology, University of Canberra, Bruce, ACT 2617, Australia; foyez.shams@canberra.edu.au; 3Conservation Horticulture, Australian National Botanic Gardens, Canberra, ACT 2601, Australia; 4Office of Nature Conservation, Environment, Planning and Sustainable Development Directorate, ACT Government, Dickson, ACT 2601, Australia

**Keywords:** conservation genetics, ex situ and in situ conservation, living collections, endangered plants, botanic gardens, dioecious

## Abstract

*Muehlenbeckia tuggeranong* is an endangered subshrub with an estimated seven individuals remaining in its native habitat, and twelve held in an ex situ living collection in the Australian National Botanic Gardens, Canberra. We conducted a genetic analysis on all known individuals of the species both in situ and ex situ to inform the conservation management of one of the rarest plants in Australia, certainly the rarest in the Australian Capital Territory. We found recent seedlings did not result from hybridisation with *M. axillaris* but resulted from sexual reproduction within the ex situ collection, leading to greater genetic diversity ex situ than in situ. However, low genetic diversity across the species indicates a high risk of extinction. Through simulations we identified the optimal breeding pairs to minimise further genetic diversity loss and increase the number of available genotypes for future reintroduction. Our work highlights the need to incorporate genetically informed breeding programs into living collections management of endangered plant species, particular those with unique life history traits.

## 1. Introduction

Small populations of endangered species are faced with extinction by numerous threats, some of which are genetic in nature [1]. Two such threats are low genetic diversity and inbreeding, which may be unavoidable in small and declining populations of species such as those listed as threatened or endangered [2,3]. Genetic diversity is viewed as particularly favourable in populations of endangered species, as high genetic diversity increases the likelihood of adaptation to changing environmental conditions [4,5], and heterozygosity is correlated with current population fitness [6,7]. Inbreeding causes losses in genetic diversity and commonly leads to inbreeding depression in small populations [6,8], which may lower fecundity and overall fitness [4]. Both low genetic diversity and the incidence of inbreeding depression increase extinction risk [9]. Even where these genetic factors are not a direct cause of extinction, most taxa are adversely affected by them prior to their extinction [10]. As such, maintaining a sufficient population size is important to reduce the incidence of inbreeding depression and retain genetic diversity [6], and preserving or enhancing the genetic diversity of critically endangered populations with compromised diversity may be necessary for species persistence [11].

A species’ life history traits will influence the genetic patterns observed within a population, and their requirements for conservation management as a result. In plant species, dioecy is one such history trait that can influence extinction risk, generally rendering plants less susceptible to inbreeding (and subsequently, inbreeding depression), as the individuals engage in obligate outcrossing [12]. However, dioecy in a small population may cause a positive feedback loop which increases extinction risk. Biased sex ratios may arise in small populations through demographic stochasticity [13], leading to reduced effective population sizes and higher inbreeding rates

Many endangered plant species are managed with living collections [14]. These may be either in situ living collections, in which plant collections are conserved in their natural habitats; or ex situ living collections, in which plants are conserved in controlled environments such as botanic gardens [14,15]. The inherent challenges in managing ex situ living collections for conservation within botanic gardens is the potential for inbreeding, diversity loss, and adaptation to cultivation [16], as well as the potential for hybridisation with closely related species which may be held by the botanic gardens [17]. This is compounded by the need to maintain accurate data and strict separation of lineages. For species facing such threats, genetic information can provide guidance on ex situ and in situ population management, allowing for the preservation and increase of a species’ genetic diversity. Genetic and demographic information can inform effective management and breeding programs for ex situ plant conservation. This may include determining how well genetic diversity in ex situ collections represents the remaining in situ populations, identifying best breeding pairs, and tracking diversity in subsequent generations [18]. Techniques such as captive breeding programs with a particular focus on the genetic requirements and risk factors of a species may offer greater conservation outcomes. For example, if a species is particularly susceptible to inbreeding due to its life history traits and population size, and has a low genetic diversity, a captive breeding program can be tailored to take these genetic risk factors and requirements into account to maximise conservation outcomes. Genetics has much to offer in the conservation of endangered species, yet as it stands, it is generally acknowledged as an overlooked and underutilised aspect of conservation [4,19], despite the role it can play in species preservation [19,20].

Genetics are a powerful tool in the conservation of endangered plant species, with applications ranging from the diagnosis of genetic factors contributing to extinction risk [21,22,23], to the design and implementation of genetically informed management strategies [24,25]. Though the goal of species preservation is ultimately the same across all conservation genetic programs, applications to manage populations of species with unique life history traits require targeted programs [26,27]. *Muehlenbeckia tuggeranong* is one such plant species, with unique reproductive traits generating additional challenges in its conservation management. The dioecious species can also reproduce vegetatively, and while the reproductive strategies can function in a complementary manner under sufficient population sizes [28], they can present additional challenges for conservation management. Using genetic tools, tailored breeding and reintroduction programs can be developed to target the challenges facing a given population, increasing the opportunity for recovery of endangered species with unique life history traits.

Here, we present the case of *Muehlenbeckia tuggeranong*, an endangered dioecious subshrub with an estimated seven individuals remaining in its native habitat along the Murrumbidgee River corridor in the Australian Capital Territory (ACT) [29,30]. Currently, an ex situ living collection of *M. tuggeranong* is held by the Australian National Botanic Gardens (ANBG), Canberra. Previous translocation efforts have been unsuccessful, and the species is at imminent risk of extinction from stochastic events. This risk of extinction is further increased by the fact that the species is dioecious, and the remaining wild population is small. Hence, conservation efforts should be approached with genetic considerations in mind. This study uses a population genetics approach to inform the conservation management of this endangered plant and highlights the use of conservation genetics in the management of in situ and ex situ populations. It aims to (1) identify the potential of hybridisation of *M. tuggeranong* with congeners held by the ANBG; (2) identify the structure, genetic diversity and level of inbreeding within the remaining wild and ex situ populations; (3) identify the best breeding pairs to maximise genetic diversity; and (4) provide advice for the conservation management of the species.

## 2. Results

### 2.1. Assessment of Potential Hybridisation and Population Genetic Structure

The ANBG seedling group was found to have high genetic similarity to the ex situ ANBG individuals (F_ST_ = 0.073: Table 1) and low genetic similarity to the other *Muehlenbeckia* species (and *Duma florulenta*), indicating that hybridisation was unlikely to have occurred. The first two PCoA axes accounted for 70% of the total variation in the dataset. *M. Tuggeranong* adult individuals formed two distinct clusters along axis 1, with each cluster associated with either male or female ex situ ANBG individuals. The clustering pattern indicates moderate genetic variation between the clusters but very little within-cluster variation (Figure 1). The wild (in situ) *M. Tuggeranong* individuals fall within the cluster on the right, containing only male ex situ individuals, suggesting the wild population is likely to contain all males. The ANBG seedlings fall directly between the male and female *M. Tuggeranong* clusters, indicating the seedlings are the result of sexual reproduction between *M. tuggeranong* individuals.

The most closely related species to *Muehlenbeckia tuggeranong, Muelenbeckia axillaris* [31], which was included in the PCoA analysis clustered along axis 2. Given that the seedlings were not clustered more closely with *M. axillaris* individuals than the *M. Tuggeranong* adults, hybridisation between the two species is considered unlikely. Further parent–offspring analysis using a pairwise zygosity comparison found mean rates of loci violating the parent offspring assumption between *M. axillaris* and ANBG seedlings (0.267) were higher than between adult *M. tuggeranong* and *ANBG seedlings* (0.0780) (Appendix A). Mean rates within species ranged from 0.0293 to 0.0734, while the mean rate between species was much higher (0.289), indicating the seedlings are of the species *M. tuggeranong* and hybridisation has not occurred.

After the possibility of hybridisation with other *Muehlenbeckia* species was excluded, SNPs were recalled from only *M. tuggeranong* individuals to maximise within-species resolution for all further analyses.

### 2.2. Analysis of Unique and Clonal Genotypes

#### 2.2.1. In Situ Population

The four wild in situ patches (A, B, C, and D) consisted of a mix of clonal and unique genotypes, with at least one unique genotype present in each patch. Of the 20 samples taken across the in situ population, 7 distinct genotypes were identified. Patches A and B each represented a single genotype and patch C contained two distinct genotypes (genotype 1: C1, C3; genotype 2: C2). The 8 samples collected from patch D represented three genotypes with six samples representing one genotype, and sample D3 and D7 each representing a unique genotype.

Five genotypes present in the wild are not held within the ANBG ex situ living collection ([A1&A2], [C1& C3], C2, D3, and D7). Genotypes [D1, D2 & D4-D8] and [B1-B7] were represented in the ex situ living collection by Clone 9 and Clone 10, respectively.

#### 2.2.2. Ex Situ Population

Five distinct genotypes were identified across the ANBG ex situ adults. The cluster of six samples (Clones 1–6) in the lower-left of the PCoA (Figure 1) represents a single female genotype while the cluster on the right of the PCoA (Clones 7–10 and the purchased plant) represents four distinct male genotypes, as Clone 9 and the purchased plant were genetically identical. Two of the identified genotypes were present in situ in patches B and D and three genotypes were only present in the ex situ collection, indicating they derived from a now-extinct source population.

### 2.3. Genetic Diversity

Genetic diversity analysis performed across the distinct genotypes identified low levels of genetic diversity across all *M. tuggeranong* management groups. The in situ wild population had the lowest observed and expected heterozygosity values (H_O_ = 0.059, H_E_ = 0.051), indicating greater genetic diversity is present in the ANBG collection than remains in the wild. The ANBG seedlings had higher observed heterozygosity (H_O_ = 0.200) than the adult ANBG plants (H_O_ = 0.135), and a lower expected heterozygosity and inbreeding coefficient (F*_IS_*) suggesting these seedlings hold more genetic diversity and are less inbred than the ANBG (adult) plants (Table 2).

Across individuals, the highest observed heterozygosity values were found in the seven ANBG seedlings and in the ANBG adult ‘Clone 1’ (Figure 2). Of the in situ individuals, the highest heterozygosity value belonged to a plant (C2) not held within the ANBG living collection.

### 2.4. Breeding Scenarios to Maximise Genetic Diversity

Genetic crosses of the 17 unique *M. tuggeranong* genotypes were performed to provide collection management recommendations to maximise the genetic diversity of the remaining population. The 17 genotypes were made up of seven seedlings, five in situ-only genotypes, three ex situ-only genotypes, and two genotypes present in both in situ and ex situ. Given that sex was known for only five individuals, simulated crosses were performed for all potential pairings, except where both individuals were known to be of the same sex, with mean observed heterozygosity across 10 simulated offspring measured. Observed heterozygosity values of the simulated offspring ranged from 0.031 to 0.480. The lowest-diversity offspring resulted from crosses between in situ individuals and ‘in situ and ex situ’ individuals, and crosses between individuals present solely in situ. The greatest diversity resulted from crosses within the ex situ (adult) group and between ex situ (adult) individuals and individuals present both in situ and ex situ (Figure 3).

## 3. Discussion

### 3.1. Management of Living Collections

Captive breeding programs and reintroductions have been shown to be successful conservation approaches for animal species [32,33], while the most common approach used for plant reintroductions is either via living collections or seed harvesting from existing populations [34,35]. However, where there is insufficient genetic diversity remaining within a species for immediate reintroduction, implementing a captive breeding-style program is likely necessary to prevent extinction. In such small populations, careful consideration must be taken in selection of founder individuals and management of breeding pairs to mitigate the risk of inbreeding and genetic drift [36,37]. Management of dioecious plant species with small population size has proven particularly challenging due to the dependence on adequate sex ratios for genetic variation [38]. The need for captive breeding protocols targeted towards plant species, particularly dioecious species, is becoming apparent, and recent studies have begun to address this gap [35,39].

Most frameworks and protocols for captive breeding and reintroductions describe broader recommendations around individual relatedness, reinforcing the need for genetically informed decision-making in species management. In many cases, for threatened plant species, living collection managers and conservation practitioners lack the resources to implement sophisticated breeding programs. To address this, we have developed a simulation approach coupled with observed heterozygosity measures to identify optimal breeding pairs, offering a simple and easily reproducible means of identifying breeding pairs without the need for more complex programs such as Colony [40].

### 3.2. Muehlenbeckia Tuggeranong Case Study

Preliminary genetic analysis of the ANBG’s ex situ living collection in 2021 suggested that there were six unique female genotypes and four unique male genotypes. However, we have identified a higher degree of clonality amongst the adult ANBG plants than expected. At present, the ANBG’s ex situ living collection holds a greater number of genetically distinct plants (genotypes) than what now occurs in the wild. The seedlings, having resulted from sexual reproduction amongst the *M. tuggeranong* adults and not from hybridization between *M. tuggeranong* and *M. axillaris*, have increased the number of unique genotypes in the living collection. We found that the in situ individuals are not at immediate risk of inbreeding depression, despite having very low genetic diversity across the population, demonstrating the benefit of vegetative propagation in preserving genetic variation [28,41].

Within the in situ population, the two wild individuals also present ex situ are known to be male; however, the sex of the other five individuals is uncertain. As no female-flowering plants have been observed amongst the four wild patches to date, and the single known female plant is genetically distinct from the males, it is speculated that there are unlikely to be any female individuals in the wild. This finding supports the expectation that the female genotype originated from a different in situ population that has since become extinct. Under this assumption, the extremely small remaining wild population of *M. tuggeranong* has lost the ability to produce new genotypes, further increasing the risk of extinction due to a lack of adaptive capacity [42]. While the relative ease with which this plant can be vegetatively propagated has allowed for genotype preservation and delayed potential inbreeding [43], the limited genetic diversity and biased sex ratios across the identified individuals of the species indicate immediate reintroduction attempts would have limited success [42,44].

The simulated crosses between individuals in different management groups provided insight on which breeding pairs would yield the highest and lowest diversity. We found that crosses with an ANBG ex situ adult parent yielded the highest diversity, primarily due to the presence of a single female individual with high observed heterozygosity. Given that this individual is the only identified female in the living collection, the risk of fertilisation with a closely related plant should be minimised by botanic gardens staff and stored with any identified females from the ANBG seedlings. Where possible, crosses within the species should be decided on an individual basis to maximise the rare opportunity for new genotypes; however, if resource limitations prevent this, crossings should be decided based on management group. Unintended crosses between closely related individuals should be avoided to ensure energetic costs for seed production are prioritised for the most viable offspring. Further sampling and analysis of the ex situ clusters are highly recommended to identify additional genotypes present, their sex and the level of genetic diversity they possess for incorporation into management and reintroduction recommendations.

Typically, genetic rescue and re-introduction involves the injection of genetic diversity into less diverse, inbred wild populations [11]. However, the scenario posed here with *M. tuggeranong* is slightly different, as whilst the remaining wild population is less diverse, it is also less inbred than the ex situ collection (in situ F_IS_ = 0.004, ex situ (adult) F_IS_ = 0.438) (Table 2). The unique reproductive traits of dioecious species with the capacity to vegetatively propagate result in a higher susceptibility to the genetic impacts of geographic isolation and small population sizes, but greater protection against genetic diversity loss, depending on clonality rates [28]. As such, conservation management of these species can be more complex, highlighting the necessity for genetically informed decision-making.

Until now, management of this species has involved collecting and maintaining propagules of established adult individuals. However, given the species’ proximity to extinction, pivoting to a targeted breeding approach will produce new genotypes while minimising the risk of inbreeding depression and reduced adaptive potential [44]. The careful selection of breeding pairs to maintain genetic diversity is a consistent recommendation across captive breeding and reintroduction frameworks, and is arguably more crucial for species with unique reproductive traits. While continuing to increase the number of genotypes held in the ANBG ex situ collection is necessary for a sufficient reintroduction pool, the expected siblingship of the seedlings and presence of only a single female adult present a challenge in maintaining genetic diversity in subsequent generations. To provide conservation management recommendations, a strong understanding of the genetic structure and variability present across populations is required.

## 4. Materials and Methods

### 4.1. Target Species

*Muehlenbeckia tuggeranong* (Tuggeranong Lignum), Family Polygonaceae, is a procumbent, sprawling subshrub that grows up to one metre high and four metres wide [29]. The species is dioecious [30], and known only from a single population occurring on flood terraces on the Murrumbidgee River corridor, south-west of Canberra in the ACT, Australia [29]. When first described in 1997, there was one female and six male plants recorded [30]. Given its small population size, very low total number of mature individuals, and restricted area of occupancy, the species is listed as *Endangered* in the ACT under the Nature Conservation Act 2014, and nationally under the Environment Protection and Biodiversity Conservation Act 1999 [45,46]. Threats to the population include flood regime, fire regime, agricultural grazing, and weed invasion [29].

The genus *Muehlenbeckia* occurs in the Southern Hemisphere, with 21 species in Australia, New Guinea, and New Zealand, and 9 in Central and Southern America [47]. The most closely related species to *M. tuggeranong*, *Muehlenbeckia axillaris* [47], occurs within similar habitats along the southern and northern tablelands of New South Wales (NSW) as well as Victoria, Tasmania, and New Zealand, with the nearest record occurring 25 km away from the known location of *M. tuggeranong* [29,30].

Following its discovery, conservation efforts for *M. tuggeranong* began as a collaboration between the ACT Government and the ANBG. On multiple occasions since the species’ discovery, cuttings were collected from all known wild plants and propagated and grown in the nursery at the ANBG. Representatives of all these collections have been maintained in subsequent years in the nursery, with careful attention paid to separation of lineages. A number of early translocation efforts failed; in one case due to an intense flood shortly after translocation. The species has never been observed to recruit in the wild, either vegetatively or non-vegetatively. Over the time since its discovery, three plants in wild populations have died. Speculated and confirmed causes of death include the 2003 bushfires, changes in river morphology, and non-targeted blackberry control measures. One plant buried in upwards of a metre of silt following flooding was presumed dead, only to later be found alive, demonstrating the plant’s resilience. The same plant has since been flooded, and it is currently uncertain whether it is still alive.

In 2021, an initial round of genetic analysis was undertaken, with the sampling of all lineages held in the ANBG (unpublished, Broadhurst. L). The results of this analysis suggested a total of 10 genets (6 female, 4 male) had been collected and held in the ANBG collection, acknowledging that some very closely related genets were defined as being distinct in the interests of maintaining any slight differences in genetic diversity. Wild plants were not sampled at the time and as such could not reliably be linked to one of the defined ex situ genets.

Despite seed-set never having been observed in wild plants, fruit of *M. tuggeranong* was observed for the first time in Feb 2022 in the ANBG nursery collection. A total of 50 seeds were harvested from numerous female plants and with germination trials in the ANBG Nursery and National Seed Bank, seeds were successfully germinated and seedlings grown on [48], proving for the first time that the species was capable of producing viable seed. Following this, subsequent seeding events have occurred each summer in the ex situ plants. At the time that the second round of genetic analysis referenced in this paper was undertaken, the ANBG ex situ collection of *M. tuggeranong* comprised 10 unique succession lines collected from the original in situ population, 7 F1-seedling genets, and a further plant found to be commercially available in local nurseries, comprising a total of 18 potential genets. The ANBG also houses plants from congener species *Muehlenbeckia axillaris*, *M. complexa*, *M. gunnii*, as well as the closely related *Duma florulenta*, raising the potential that hybridisation was the basis of seed formation in the ex situ *M. tuggeranong* plants.

### 4.2. DNA Extraction, Amplification, Sequencing and Genotyping

Young leaves were collected and dried in silica gel in preparation for extraction from all known individuals of *Muehlenbeckia tuggeranong.* Twenty wild samples were taken across four patches of varying sizes. Prior to genetic testing, it was assumed that these four patches made up four individual plants. Within each of the four patches, samples were taken approximately 50 cm apart. The number of samples taken per patch was dependent upon the patch size. Two samples were taken in patch A, seven in patch B, three in patch C, and eight in patch D. The patches were a minimum distance of 75 metres and maximum distance of 980 m apart, with an average distance of 405 metres apart. Metadata, including spatial and photographic records, were taken for each sample. Leaves were also sampled from 18 ex situ individuals held in the ANBG living collection, which included ten unique succession lines grown from cuttings collected from wild populations since the discovery of the species (suspected at the time of sampling to be six unique females and four unique males), one plant available for purchase from local nurseries (male), and seven seedlings grown from seed collected from the ANBG adult plants. Ex situ plants from the ANBG collection across *Muehlenbeckia axillaris* (3), *Muehlenbeckia complexa* (2) and *Muehlenbeckia gunnii* (2) and *Duma florulenta* (2) were also included to investigate potential hybridisation.

DNA sequencing was completed using the DArTseq^TM^ (University of Canberra, Bruce, ACT 2617, Australia) reduced-representation sequencing approach [49,50]. Filtering was performed using the R package suite ‘dartRverse’ [49,50]. and ‘adegenet’ [51]. Individuals with call rates below 25% were removed, along with loci with reproducibility below 99%, and read depths outside the range 5–500. Finally, loci with call rates below 97%, monomorphic loci, and secondaries were removed, resulting in 1271 SNPs with 1.07% missing data. Once potential hybridisation was eliminated, SNPs were recalled from only *M. tuggeranong* populations. The above process was followed for individual call rate, reproducibility, and read depth filtering. Loci with call rates below 95%, monomorphic loci, and secondaries were then removed, leaving 1128 SNPs with 2.29% missing data.

### 4.3. Genetic Analysis

All genetic analyses were performed using the ‘dartRverse’ package suite within the program R unless otherwise stated [49,50]. Relatedness between management groups of *Muehlenbeckia tuggeranong*, *Duma florulenta* and other *Muehlenbeckia* species held by the ANBG (*M. axillaris*, *M. gunnii*, *M. complexa*) was analysed to identify the parental group of ANBG seedlings, given the potential for interspecific hybridisation within the ANBG living collection. Pairwise F_ST_ was used to assess the potential for hybridisation in seedlings. To avoid sample-size bias, *M. tuggeranong* management groups were iteratively subsampled 1000 times and F_ST_ values generated, with mean F_ST_ reported. A principal component analysis (PCoA) provided visualisation of genetic differences between management groups (in situ, ex situ adults, and ex situ seedlings), and the closest related congener (*M. axillaris*). A pairwise zygosity comparison [52] was used to verify potential hybridisation between the *M. axillaris* and *M. tuggeranong* individuals held within the ANBG living collection facility. This approach compared loci between each seedling and parent combination across the two species, identifying the proportion of loci that violated the assumption of a parent–offspring relationship. For instance, an individual homozygous for allele A at a given locus would not produce an offspring homozygous for allele B, this scenario violates the parent–offspring assumption.

To determine genetic diversity and structure within and between management groups of *M. tuggeranong*, levels of genetic diversity within management groups were assessed using observed and expected heterozygosity values and the inbreeding coefficient F_IS_. Pairwise F_ST_ was used to determine relatedness between management groups. Principal component analysis (PCoA) provided visualisation of genetic differences across individuals and management groups.

To identify the number of unique genotypes amongst in situ and ex situ populations, potential clones and parent–offspring relationships within the dataset, a likelihood-based method for pedigree reconstruction was used without any priori information on potential parents. The analysis was performed using COLONY v2.0.7.0 software [40].

To provide future management recommendations, all potential genetic crosses between the 17 unique genotypes were simulated using the gl.sim.crosses function in the ‘dartRverse’ package [49,50]. Crosses where parents were known to be of the same sex were not simulated. For each cross, 10 offspring were simulated to reflect the approximate seed yield of the species, and the average heterozygosity across the offspring was calculated.

## 5. Conclusions

We have identified that existing conservation management of endangered plant species through seed banks and living collections may not be sufficient to prevent extinction of in situ populations. In doing so, we have emphasised the need to incorporate targeted breeding programs into collections management of endangered plants, both to preserve existing genotypes and to improve the overall genetic fitness of ex situ populations. This strategy is particularly important where species possess unique life history traits, as with the dioecious *M. tuggeranong*. These programs maximise the rare opportunity to increase the genetic variability within a collection, improving the likelihood of reintroduction success and species persistence. Such programs are common practice in zoos and conservation parks for bolstering genetic fitness of threatened fauna populations and are widely applied in managing vital crop species to ensure food security [53,54,55]. Incorporating genetically informed breeding programs into seed banks offers a promising two-fold approach that will allow us to both maintain and increase genotypes [56].

## Figures and Tables

**Figure 1 plants-14-01812-f001:**
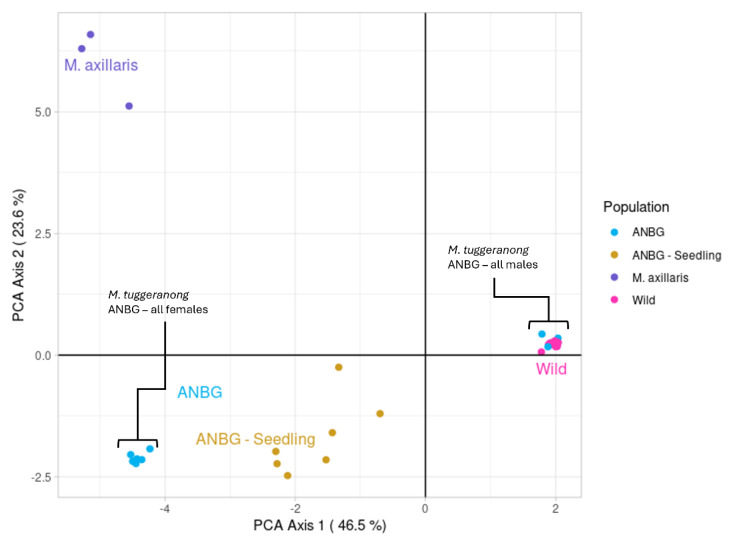
Plot from PCoA analysis on the wild, ANBG adult, and ANBG seedling genotypes and the closely related congener species *M. axillaris* from PCoA analysis on the wild, ANBG adult, and ANBG seedling genotypes and the closely related congener species *M. axillaris*.

**Figure 2 plants-14-01812-f002:**
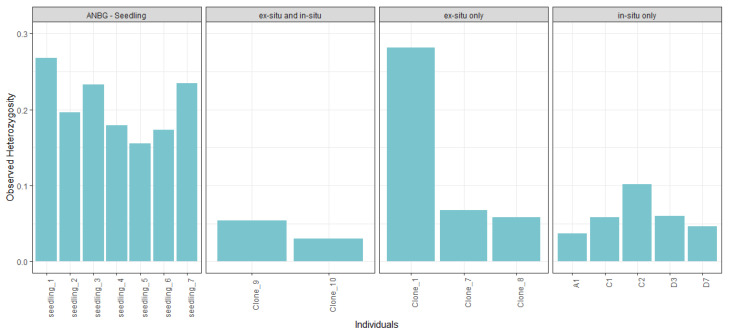
Observed heterozygosity values of unique individuals, grouped by seedlings (far **left**), adults present both ex situ and in situ (middle **left**), adults present in ex situ only (middle **right**) and adults present in situ only (far **right**).

**Figure 3 plants-14-01812-f003:**
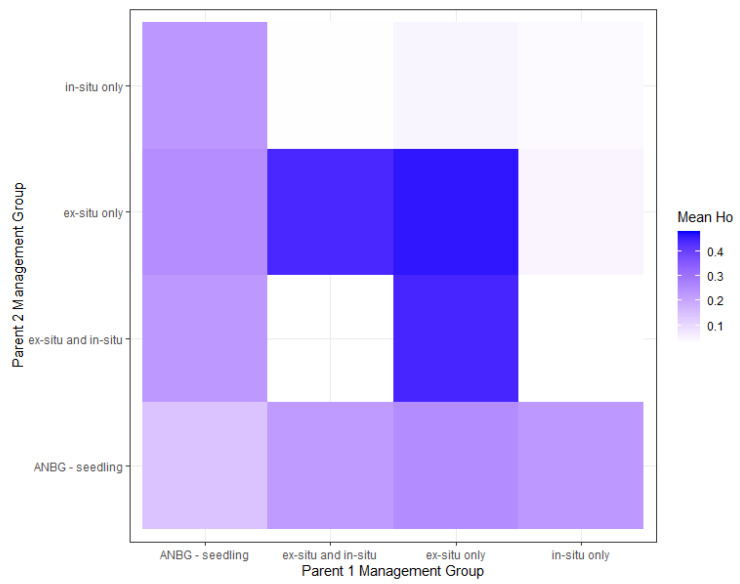
Heat map of mean observed heterozygosity values derived from simulated crossings between individuals; summaries organized by management group.

**Table 1 plants-14-01812-t001:** Pairwise FST values for the *Muelenbeckia tuggeranong* held in the ANBG as the adult and seedling populations, the wild population, and three other *Muelenbeckia* housed in the ANBG collections. FST values derived after subsampling from *M. tuggeranong* groups for 3 individuals (targeting comparison with *M. axillaris*), and taken as the mean FST over 1000 subsampling iterations.

	ANBG—Seedling	ANBG	Wild	*M. axillaris*	*M. gunnii*
ANBG (n = 12)	0.073	NA	NA	NA	NA
Wild (n = 21)	0.373	0.171	NA	NA	NA
*M. axillaris* (n = 3)	0.501	0.452	0.629	NA	NA
*M. gunnii* (n = 1)	0.905	0.897	0.941	0.875	NA
*M. complexa* (n = 2)	0.885	0.867	0.950	0.836	0.936

**Table 2 plants-14-01812-t002:** Observed (HO) and expected (HE) heterozygosity and inbreeding coefficient (FIS) for the ANBG ex situ seedlings and ANBG adult plants and wild in situ plants.

Management Group	HO	HE	FIS
ANBG seedlings	0.200	0.233	0.198
ANBG	0.135	0.244	0.438
Wild	0.059	0.051	0.004

## Data Availability

The original data presented in the study are openly available in Digital Commons Data, under the https://doi.org/10.5281/zenodo.15645088.

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
