# Peer review of "On the Precipice of Extinction: Genetic Data in the Conservation Management of In Situ and Ex Situ Collections of the Critically Endangered Muehlenbeckia tuggeranong (Tuggeranong Lignum)"

_plants, 2025, doi:10.3390/plants14121812_

Round 1
Reviewer 1 Report
Comments and Suggestions for Authors The authors conducted genetic analyses of both in-situ and ex-situ populations of the endagered species Muehlenbeckia tuggeranong. They distinguished genotypes within each population and found that the ex-situ collection exhibited greater genetic diversity but also higher levels of inbreeding compared to the in-situ population. Additionally, their analyses indicated that the ex-situ individuals are unlikely to have hybridized with closely related species. Simulation studies further demonstrated that certain mating patterns could maximize genetic diversity. In this study, all available individuals of the plant species were analyzed to enable comprehensive identification of distinct genotypes. The methodology employed for genetic analysis is appropriate, and the interpretation of the resulting data is both sound and scientifically rigorous. Figures and tables are clearly presented and effectively support the findings. Overall, the study provides valuable and practical data that could meaningfully inform genetic rescue efforts in wild Muehlenbeckia tuggeranong populations. I consider this work highly suitable for publication in this journal.Author Response
Comments 1: The authors conducted genetic analyses of both in-situ and ex-situ populations of the endagered species Muehlenbeckia tuggeranong. They distinguished genotypes within each population and found that the ex-situ collection exhibited greater genetic diversity but also higher levels of inbreeding compared to the in-situ population. Additionally, their analyses indicated that the ex-situ individuals are unlikely to have hybridized with closely related species. Simulation studies further demonstrated that certain mating patterns could maximize genetic diversity. In this study, all available individuals of the plant species were analyzed to enable comprehensive identification of distinct genotypes. The methodology employed for genetic analysis is appropriate, and the interpretation of the resulting data is both sound and scientifically rigorous. Figures and tables are clearly presented and effectively support the findings. Overall, the study provides valuable and practical data that could meaningfully inform genetic rescue efforts in wild Muehlenbeckia tuggeranong populations. I consider this work highly suitable for publication in this journal
Response 1: We appreciate the feedback provided by this reviewer. The reviewer made no recommendations for revisions and as such we have no changes corresponding directly to this feedback. We thank the reviewer for their time.
Reviewer 2 Report
Comments and Suggestions for Authors
The ms characterize the genetic diversity of a rare (the rarest) species in Australia, analyzing in situ (10 individuals) and ex situ (20 individuals) collections. This species was published for the first time in 1997. They found that ex situ collections are more diverse genetically. The objetive is to inform results to conservation management office about the best breeding pairs to maximise genetic diversity and give recommendations based on life history traits.
The ms gives important information about the species Muehlenbeckia tuggeranong but I have some comments.
In a recent phylogenetic study Muehlenbeckia tuggeranong is sister to M axillaris and the first description of this rare species states that it is very closely related based on morphological characteristics, being the differences in the terminal inflorescence while M. axillaris is polymorphic for that trait.
It was proposed that Muehlenbeckia tuggeranong reached Australis though dispersal and that was dated quite recently (Schuster et al 2013). So the species has suffered genetic drift due to founder effect. It is interesting to follow the population to see how it evolves.
Maybe a comparison of genetic diversity of this two species gives relative information on genetic diversity in the lineage (closely related species) and gives more information on how unique is this rare species.
The FST values shown in table 1 for the other species is based on how many individuals? The authors only mention the in situ (20 samples) and the ex situ (18 samples) but they don’t mention how many of the other species. This is relevant not only to detect hybridization but to compare genetic diversity indices.
I think it is important to compare genetically M. tuggeranong and M axillaris they are very similar and phylogenetically is its sister species (Desjardins et al.2023). More sampling of wild populations of M. axillaris is needed to do a more inclusive analyses to see if M. tuggeranong falls within the variation of M axillaris. They show the PCA, where in situ + ex situ males, ex situ females and M. axillaris form each an individual group but this is based on small sample size and can be biased, It is strange also that in situ and ex situ form individual groups since the ex situ plants are from the wild plants, they are the same population, and not that much time has passed to differentiate; this is evidenced by the small Fst value shown in table 1 and also that the male plants from ANBG are cluster within the wild individuals.
The wild population is mostly male plants maybe that is why in the PCA they cluster with ex situ male plants.
The observed heterozygosity value for the wild individuals is smaller than for ex situ individuals; however, it is similar to expected heterozygosity. Instead for the ex situ individuals the expected heterozygosity is almost twice the observed heterozygosity indicating they are not in equilibrium. That is why both FIS differ indicating inbreeding in ex situ. The authors should discuss more these findings. Also comparison to closely related species would be good to see how diverse are species within the lineage.
Page 127 line 116 should say species instead of closely related genus,
Line 214 I don’t understand were this affirmation comes from "These ANBG plants hold up to 7 times more genetic diversity than remains in the wild population, highlighting the value offered to the conservation of this endangered species by the ANBG ex-situ living collection."
Mentioned References:
Schuster, T. M., Setaro, S. D., & Kron, K. A. (2013). Age estimates for the buckwheat family Polygonaceae based on sequence data calibrated by fossils and with a focus on the Amphi-Pacific Muehlenbeckia. PLoS One, 8(4), e61261.
Desjardins, S. D., Bailey, J. P., Zhang, B., Zhao, K., & Schwarzacher, T. (2023). New insights into the phylogenetic relationships of Japanese knotweed (Reynoutriajaponica) and allied taxa in subtribe Reynoutriinae (Polygonaceae). PhytoKeys, 220, 83.
Reviewer 3 Report
Comments and Suggestions for Authors
This paper focuses on the genetic analysis of Muehlenbeckia tuggeranong, an endangered dioecious subshrub native to the Australian Capital Territory. With fewer individuals remaining in the wild, the species faces significant conservation challenges. The study examines both in-situ and ex-situ populations to assess genetic diversity, structure, and the potential for hybridisation with related species. By identifying unique genotypes and assessing breeding strategies, the research aims to inform conservation management decisions and support the long-term survival of this rare plant species.
Abstract
The phrase "fewer than ten individuals remaining in its native habitat" could be more specific about the exact number or a clear range if possible. This will strengthen the impact of the statement on the reader.
The abstract could benefit from providing a brief explanation of why the findings about genetic diversity (ex-situ vs. in-situ) are significant in terms of long-term conservation.
Introduction
The phrase "inherently greater or lower susceptibility to the risks" could be simplified for clarity and flow. Consider rewording it to focus more directly on the core idea.
The introduction could benefit from clearer transitions when shifting from general discussions of genetic threats to specifics about the species in focus. This would help to more smoothly guide the reader to the core of the study.
Results
The section would benefit from a more explicit summary or interpretation of key results from the PCoA analysis in the text, particularly how the clustering patterns relate to the main objectives of the study.
In section 2.2, consider simplifying the explanation about clonal genotypes to make it more reader-friendly, as the information about which genotypes are represented where is complex.
Discussion
The paragraph discussing the clonality among the ANBG plants could benefit from a clearer statement about the significance of these findings in the context of the species’ overall genetic health and conservation.
When discussing reintroduction and genetic management, it would strengthen the argument to emphasize the practical steps for improving genetic diversity in future reintroductions more directly.
Conclusion
I suggest the author add a conclusion section.
Round 2
Reviewer 2 Report
Comments and Suggestions for Authors
I think the ms has been improved from previous version, I have. no further comments.